# Uncovering the chiral bias of meteoritic isovaline through asymmetric photochemistry

Jana Bocková [1], Nykola C. Jones [2], Jérémie Topin [1], Søren V. Hoffmann [2] & Cornelia Meinert [1] ✉

Systematic enrichments of L-amino acids in meteorites is a strong indication that biological homochirality originated beyond Earth. Although still unresolved, stellar UV circularly polarized light (CPL) is the leading hypothesis to have caused the symmetry breaking in space. This involves the differential absorption of left- and right-CPL, a phenomenon called circular dichroism, which enables chiral discrimination. Here we unveil coherent chiroptical spectra of thin films of isovaline enantiomers, the first step towards asymmetric photolysis experiments using a tunable laser set-up. As analogues to amino acids adsorbed on interstellar dust grains, CPL-helicity dependent enantiomeric excesses of up to 2% were generated in isotropic racemic films of isovaline. The low efficiency of chirality transfer from broadband CPL to isovaline could explain why its enantiomeric excess is not detected in the most pristine chondrites. Notwithstanding, small, yet consistent L-biases induced by stellar CPL would have been crucial for its amplification during aqueous alteration of meteorite parent bodies.

Chirality, or handedness, is an important property in nature that plays a central role in regulating biochemical reactions[1]. Since life evolved as homochiral, stereospecific interactions are crucial for maintaining the proper function of biological systems and are involved in many biological processes, such as molecular recognition, protein folding, cellular signaling, and enzyme specificity[2–4]. However, the origins of the universal predominance of L-amino acids in proteins and D-sugars in nucleic acids remain to be elucidated. Given that meteoritic and terrestrial proteinogenic amino acids share the same handedness of enantiomeric excess (*ee*)[5], the delivery of enantioenriched chiral organics via Solar System debris[6,7] may have driven the evolution of biological homochirality by biasing the initially racemic prebiotic pool on the early Earth.

The non-proteinogenic amino acid isovaline (Fig. 1a) stands out amongst most extra-terrestrial chiral organics due to its presence in large L-enantiomeric excess (*ee*L) of up to about 20% in a number of carbon-rich meteorites, so called carbonaceous chondrites[8–10]. Given

that isovaline is mostly absent in the Earth's biosphere except for a few fungal peptides containing predominantly the D-form[11], the molecule represents a robust test case in the quest for chiral excess in extraterrestrial samples. Moreover, this α-methyl-amino acid has greater resistance to racemization over geological timescales, and thus has greater potential to preserve its chiral bias compared to proteinogenic α-hydrogen-amino acids. This has long been considered as the explanation of its relatively high *ee* content in several carbonaceous chondrites[9,12–15]. Interestingly, the magnitude of *ee*L of isovaline shows a positive correlation with the extent of aqueous alteration in CI, CM, and CR chondrites (chondritic classes petrologically and compositionally similar to the Mighei, Ivuna, and Renazzo meteorites, respectively)[10,16–18], while it appears inversely related to its overall abundance[5,16,19], and demonstrates yet another unique property of meteorites. Even though the impact of aqueous alteration caused by melted water ice on the inorganic mineral content of parent bodies is rather well understood, the underlying mechanisms that explain how

[1]Institut de Chimie de Nice (ICN), CNRS UMR 7272, Université Côte d'Azur, 06108 Nice, France. [2]ISA, Department of Physics and Astronomy, Aarhus University, 8000 Aarhus C, Denmark. ✉e-mail: cornelia.meinert@univ-cotedazur.fr

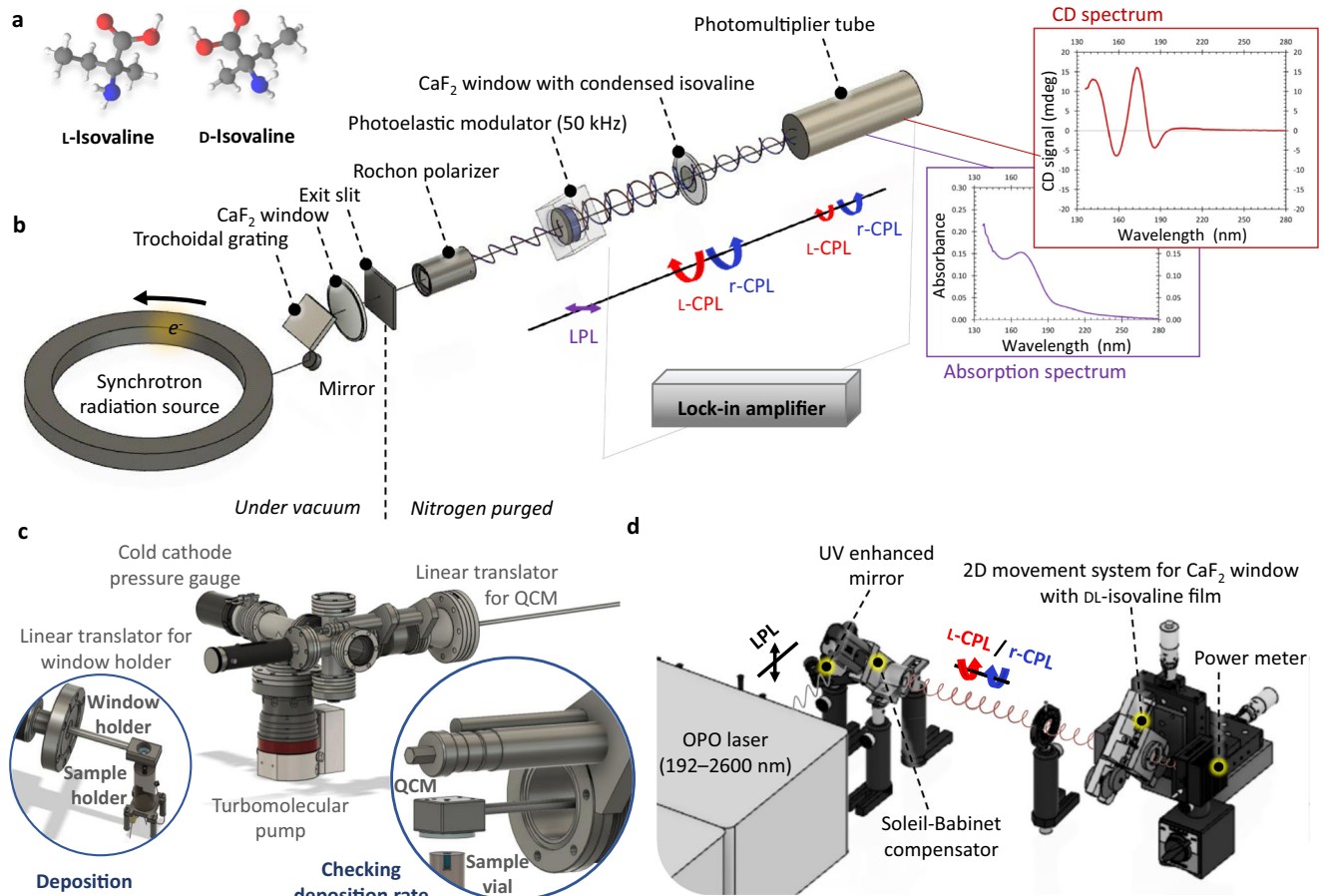

**Fig. 1 | Experimental observation of polarization controllable symmetry breaking in analogs mimicking isovaline adsorbed on interstellar dust grains. a** Schematic structure of the L- and D-enantiomers of isovaline (carbon in gray, hydrogen in white, oxygen in red, and nitrogen in blue). **b** Scheme of the AU-CD beam line apparatus at the ASTRID2 synchrotron storage ring facility, ISA, Aarhus University, Denmark, used for recording UV circular dichroism (CD) and anisotropy spectra of L- and D-isovaline thin films deposited on UV grade $CaF_2$ windows. A $CaF_2$ photoelastic modulator converts horizontally polarized synchrotron radiation into 50 kHz alternating left- and right-circularly polarized light (L- and r-CPL). After passing through enantiopure isovaline films, the transmitted light is recorded using a vacuum UV enhanced photomultiplier. **c** Scheme of the sublimation-deposition chamber for preparation of racemic thin films of isovaline for asymmetric photolysis experiments. The films are produced by condensation of sublimated DL-isovaline powder on a $CaF_2$ window. Their thickness is monitored by a quartz crystal microbalance (QCM). **d** Scheme of the tunable laser set-up employed for asymmetric photolysis experiments on racemic thin films of isovaline. Linearly polarized monochromatic laser radiation (LPL) is reflected by a UV enhanced mirror and is subsequently circularly polarized by a Soleil-Babinet compensator. Monochromatic UV CPL then interacts with a racemic DL-isovaline film condensed on a $CaF_2$ window. The window constantly moves in the xy direction to allow for homogeneous sample irradiation.

these geological processes would have affected the *ee* are not yet fully elucidated, despite their significant relevance for chiral amplification.

Based on a growing body of evidence, chiral light-matter interactions have been so far considered as one of the most likely triggers of the observed asymmetry in meteoritic amino acids[20]. However, this hypothesis has yet to be proven conclusively. Notably, laboratory simulations employing ultraviolet circularly polarized light (UV CPL) were shown to induce *ee* of a few % in racemic alanine[21] and leucine[22,23], as well as in a total of five amino acids synthesized from initially achiral precursors in inter-/circumstellar ice analogs[24,25]. In this context, near-infrared (NIR) polarimetry observations revealed large areas of CPL in the Orion[26] and NGC 6334V[27] star-forming regions attributed to the presence of partially aligned spheroidal dust grains[28–30]. Scattering from such aligned grains and dichroic extinction that are likely involved in the production of IR CPL in space[28–30], are also capable of producing circular polarization in the UV, which is necessary to drive asymmetric photochemistry[29].

The asymmetric interactions of UV CPL with chiral species stems from different absorption cross-sections of enantiomers when irradiated with left- or right-CPL (L-CPL or r-CPL), an effect called circular dichroism. The fractional difference in the absorption ($\Delta\varepsilon/\varepsilon$, with $\Delta\varepsilon$ being the differential extinction and $\varepsilon$ the extinction coefficient) is given by the anisotropy factor $g$, which as derived by Kagan et al.[31]. (Supplementary Note 1) directly reflects on the *ee* inducible by preferential UV-photon induced fragmentation of the more absorbing enantiomer at a given wavelength and helicity of CPL. Since the structure, and hence the chiroptical response, of enantiomers critically depends on their immediate surroundings[32–34], investigating the anisotropy spectra of chiral organics in different environments representing astrochemical conditions is essential. The chiroptical properties of isovaline have been previously successfully measured in aqueous solution[33] mimicking molecules embedded in water-rich interstellar ices[35], as well as in the gas-phase simulating molecules desorbed in so-called interstellar dust cycles[36]. In contrast, the elucidation of the asymmetric absorption of UV CPL by isovaline enantiomers adsorbed on the surface of interstellar dust grains has been so far limited only to a few inconsistent results[37–40]. This is somewhat unsurprising since experimental solid-phase chiroptical spectroscopy remains highly challenging because of the artefacts that result from macroscopic anisotropies, such as linear dichroism and linear

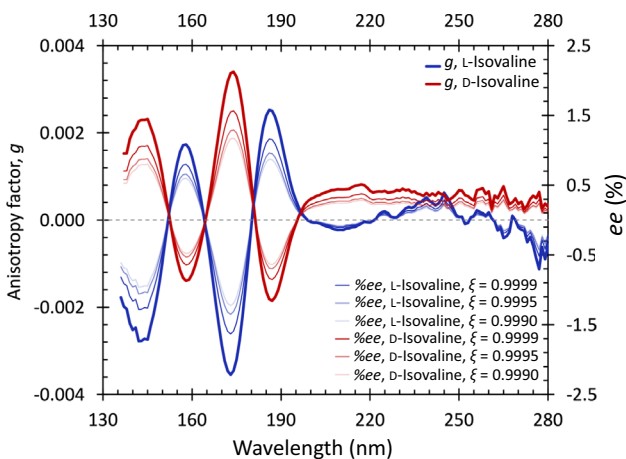

**Fig. 2 | Chiroptical solid-phase spectra of isovaline.** The anisotropy spectra (thick lines) of the L- and D-enantiomers of isovaline allow for the prediction of enantiomeric excess (*ee*) inducible by asymmetric photolysis by circularly polarized light at the extent of reaction $\xi$ (thin lines of decreasing intensity at 0.9999, 0.9995, and 0.9990) via numerical solution of the equation reported by Kagan et al.[31]. Source data are provided as a Source Data file.

birefringence, as well as from scattering and absorption flattening[41,42]. Overcoming these has far-reaching implications beyond the study of homochirality due to the widespread applications of chiroptical-property-bearing thin films in quantum optics, biophotonics, chiroptoelectronics, or life sciences[43].

Isovaline's relatively high L/D ratios, consistently found in heavily aqueously altered meteorites are extensively discussed in the context of amplification during parent body aqueous processing. However, the role of asymmetric photochemistry capable of inducing the necessary initial L-bias in meteoritic isovaline to direct the amplification, remains poorly defined and experimental data is lacking. By using our extensive experience in circular dichroism and anisotropy spectroscopy for thin films, we were able to unravel the genuine chiroptical properties of solid-phase isovaline (Fig. 1b). Strengthened by asymmetric photolysis experiments utilizing circularly polarized UV laser radiation (Fig. 1c, d), we confirmed the helicity-dependent enantioselectivity of CPL interacting with this non-proteinogenic amino acid. Our new data can finally account for the racemic or nearly racemic composition of isovaline in the most pristine, least altered, carbonaceous chondrites such as CM2.7 Paris[44] and CM2.6–2.0 Winchcombe[19] compared to the high L-excess found in the most aqueously altered ones without violating the hypothesized astrophysical CPL scenario. The present findings are therefore critical for advancing the search for chiral biosignatures in outer space by guiding future enantioselective analyses of extraterrestrial samples.

## Results

### Solid-phase circular dichroism and anisotropy spectroscopy

To record accurate anisotropy spectra of thin films of isovaline enantiomers in the VUV/UV spectral range (130–330 nm), differential absorption (CD) and absorption was measured simultaneously[45] at the AU-CD beamline of the ASTRID2 synchrotron storage ring facility, ISA, Aarhus University, Denmark. A detailed description of the sample preparation procedure and the spectroscopic measurements is in the Methods section, and the corresponding CD and absorbance spectra are shown in Supplementary Fig. 1. The anisotropy (*g*) spectrum of L-isovaline (Fig. 2) exhibits four well resolved bands with minima at 142 nm and 173 nm, and maxima at 158 nm and 186 nm. The anisotropy signal of the less optically pure D-enantiomer quasi-perfectly mirrors the one of its optical antipode, confirming the overall consistency of the present data set. Employing the equation reported by Kagan et al.[31],

the measured *g* values allow to derive the *ee* which can be generated by asymmetric photolysis of a racemic mixture with monochromatic CPL at a given extent of photolysis $\xi$ (Fig. 2). The *g* spectra are less reliable above ~198 nm due to low CD and absorbance signals in combination with minor distortions induced by residual scattering effects (Supplementary Note 2, Supplementary Fig. 2). These also account for slight discrepancies in the positions and intensities of the anisotropy bands in Fig. 2.

To our knowledge, the only other anisotropy spectrum of solid-phase isovaline in the literature is our own study from 2012[38], where significant contribution of scattering distorted the true anisotropy signals (for a detailed comparison and discussion see the Supplementary Note 2). Two CD spectra of solid-phase isovaline reported by other groups are available in the literature. An experimental spectrum[37] valid solely down to 180 nm due to absorption saturation at shorter wavelengths, and a theoretical one[40] showing bands with extremely low rotational strengths of ambiguous sign and spectral position, thence hindering any meaningful comparisons. The accuracy of the present data set is strengthened by the apparent similarity of the CD/anisotropy spectra of L-isovaline with the ones previously reported for the structurally related amino acids L-alanine and L-valine[38,39,46]. Essentially, the major dichroic transitions coincide, while the exact spectral positions and intensities of the corresponding bands are shaped by the intra- and intermolecular perturbations of the specific alkyl side chains and/or α-substituents. Based on the analogy with L-alanine and L-valine[46], the anisotropy band of L-isovaline with a maximum at 186 nm is likely to be due to the $\pi_0 \to \pi^*$ transition of the carboxylate anion, the band with a minimum at 173 nm due to the $n(COO^-) \to \sigma^*(N-H)$ transition, and the band with a maximum at 158 nm can be tentatively ascribed to the $\pi_1(COO^-) \to \sigma^*(N-H)$ transition mixed with the $n \to \pi^*$ transition of the carboxylate anion. The prominent *g* band in the spectra of L-alanine and L-valine with a minimum just above 200 nm, originating from the $n(COO^-) \to \sigma^*(N-H)$ transition mixed with the $n \to \pi^*$ transition of the carboxylate anion[46], is much more suppressed in solid-phase isovaline, as it is the case in aqueous solution[47]. Hence, this band is not clearly distinguished in Fig. 2 due to the low signals in combination with the distortions by residual scattering.

Theoretical calculations based on time-dependent density functional theory (TDDFT) were performed to reproduce the conformation-specific CD bands. Supplementary Fig. 5 shows the theoretical CD spectra in which the rotatory strengths (*R*) of the first 200 excited states were calculated at the B3P86/6-311 + G(d,p) level of theory. Of the four conformers considered (Supplementary Data 1), only conformer IIa provided good agreement with the experimental CD spectrum. The most energetically favored isovaline conformation according to our TDDFT calculations did not show the strongest correlation with the experiment, which agrees with previous observations for isotropic films of zwitterionic alanine[21]. This suggests that conformer IIa may be the dominant conformation of isovaline in the amorphous solid state. The first coherent experimental chiroptical spectra of thin films of isovaline enantiomers reported here therefore extend the frontiers of theoretical circular dichroism to further confirm the nature of the underlying dichroic transitions.

### Asymmetric photolysis of racemic isovaline films

To experimentally confirm the energy-selective helicity-dependent photolysis of racemic solid-phase isovaline, a new experimental set-up was constructed. It comprises a high-vacuum sublimation-deposition chamber for thickness-controlled thin film preparation (Fig. 1c) and a tunable laser set-up for UV CPL irradiation (Fig. 1d). For the asymmetric photolysis experiments, four sets of racemic ~500 nm thick isovaline films were prepared. One film from each set was irradiated either with L- or r-circularly polarized laser radiation, while the other one remained non-irradiated to serve as a reference for post-irradiation analyses by multidimensional gas-chromatography coupled to reflectron

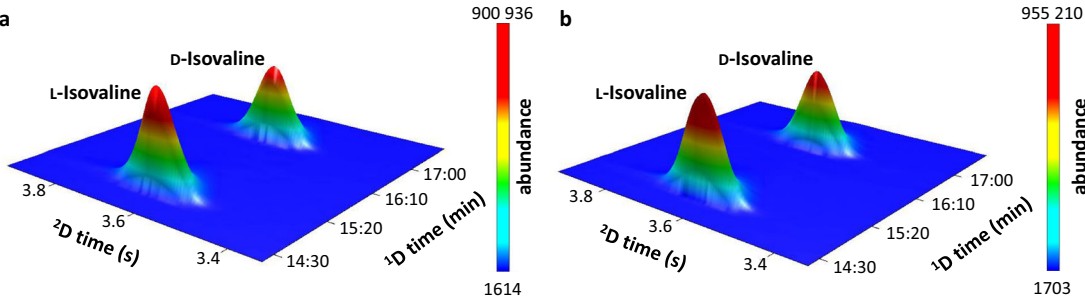

**Fig. 3 | Comparing enantiomeric excesses in irradiated versus non-irradiated isovaline films. a** Enantioselective multidimensional gas-chromatography coupled to time-of-flight mass-spectrometry (GC×GC–TOF-MS) ion chromatogram at $m/z$ 168 corresponding to the L- and D-enantiomers of isovaline in the r-CPL irradiated film. **b** GC×GC–TOF-MS ion chromatogram at $m/z$ 168 of the non-irradiated diluted (1:600) reference sample (sample set IV). Note that ~1% $ee$ difference of the irradiated film compared to the non-irradiated counterpart cannot be readily visible in the 3D plots.

**Table 1 | Opposite handedness of the enantiomeric excess (ee) induced in racemic isovaline films in asymmetric photolysis experiments by 192 nm left (L-) and right circularly polarized light (r-CPL) confirms the predictions by the anisotropy spectra**

| Sample set | CPL helicity | Irradiation time | Number of replicate GC×GC injections, $n_{irr}$ /$n_{non\text{-}irr}$ | $\%ee_L \pm$ SD | Two-sample $t$ test, $p$-value (two-tailed) | $\%ee_L$ predicted[a] at $\xi = 0.9999$ |
|---|---|---|---|---|---|---|
| I | L-CPL | 14 h | 9 / 16 | $-2.05 \pm 1.05$ | $4.8 \times 10^{-6}$ | $\leq -0.6\%$ |
| II | L-CPL | 9 h 35 min | 9 / 9 | $-1.02 \pm 0.85$ | $2.3 \times 10^{-3}$ | |
| III | r-CPL | 8 h 15 min | 9 / 10 | $1.89 \pm 1.08$ | $4.1 \times 10^{-5}$ | $\geq 0.6\%$ |
| IV | r-CPL | 6 h 30 min | 9 / 9 | $1.27 \pm 0.51$ | $1.4 \times 10^{-6}$ | |

[a]Based on the $g$ values of the L-enantiomer of isovaline using the equation reported by Kagan et al.[31]. The L-enantiomer was selected due to its higher purity compared to the D-enantiomer (see Methods).

time-of-flight mass spectrometry (GC×GC–TOF-MS). Note that chromatographic techniques do not necessarily allow to measure absolute $ee$ values accurately[48]. Similar concentrations, signal intensities of irradiated and corresponding diluted non-irradiated sample (Fig. 3) within each sample set were important to accurately determine the absolute difference in $ee$-s, and hence reliably assess the effect of CPL on the enantiomeric excess excluding any potential instrumental artefacts[48].

To probe the chiral selection by asymmetric photochemistry associated with the $\pi_0 \rightarrow \pi^*$ transition of the carboxylate anion, the shortest radiation wavelength of our laser, 192 nm, was selected in the present study. Based on the kinetics described by Kagan et al.[31], the predicted $|\%ee|$ at $\xi = 0.9999$ for the more optically pure L-enantiomer with a $g_{192} \geq 0.0012$ is ~0.6%. The residual scattering effects present in our anisotropy spectroscopy measurements here would, however, falsely increase absorbance and hence artificially lower the calculated anisotropy factors. Therefore, 0.6% at $\xi = 0.9999$ can be solely understood as the lower limit estimate. Notwithstanding, during the stereoselective photolysis by 192 nm CPL, the $ee$ of isovaline scales[38] with the progressing extent of reaction $\xi$ (for $\xi$ close to 1) as ~ $(1-(1-\xi)^{|g|/2})$ and consequently, very high photolysis rates are necessary to achieve reliably measurable $ee$-s. These are, however, associated with low analyte quantities for post-irradiation analyses. To overcome this analytical challenge, GC×GC–TOF-MS analysis of $N$-trifluoroacetyl-$O$-methyl ester derivatives of isovaline was selected due to its high enantio-resolution power (Fig. 3, $R_S$ = 4.7) and detectability[48]. Two replicate irradiations of racemic isovaline films by 192 nm L-CPL resulted in the D-excess of isovaline of up to ~2% as opposed to their non-irradiated counterparts (Table 1). The opposite effect, i.e. L-excess, was observed for two replicates irradiated by r-CPL. Two-sample Student's $t$ tests confirmed the statistical significance of the difference in the $ee$ values of the irradiated and non-irradiated films of each set (Table 1 and Supplementary Tables 1, 2). The details of the statistical analysis are in the Methods section. The handedness and magnitude of the enantiomeric excess induced by asymmetric photolysis of racemic isovaline films by 192 nm circularly polarized laser radiation, therefore,

generally agrees with the predictions of the present anisotropy spectra with an extremum at ~186 nm. While this result further supports the accuracy of the anisotropy spectra in Fig. 2, the effect of asymmetric photolysis by more astrophysically relevant broadband stellar CPL and the implications for the CPL scenario are elaborated in the Discussion section.

## Discussion

The systematic excess of L-amino acids in carbonaceous chondrites has remained an unresolved riddle to date. Pointing to the potential significance of the extra-terrestrial matter at the dawn of biological homochirality, efforts to its resolution have sparked extensive research[5]. The non-proteinogenic nature of isovaline largely dispels any doubts about potential terrestrial sources of its chondritic L-excess[9,12–15], making isovaline a kingpin molecule in this scientific endeavor. Despite its presence presumably in the L-excess[5] on the early Earth, isovaline did not make it to the 20 proteinogenic amino acids. However, thanks to its relative resistance to racemization due to the protecting α-methyl substituent, L-enriched isovaline is likely to have shaped the evolution of biological homochirality of both amino acids and sugars. For amino acids, this possibly occurred via decarboxylative transamination reactions favoring the synthesis of L-α-hydrogen-amino acids[49] or via facilitating the assembly of chiral secondary structures[15]. As for the homochirality of sugars, isovaline was found to have an asymmetric catalytic effect on the synthesis of threose and erythrose[12].

The general observation of the $ee_L$ of isovaline scaling with the extent of aqueous alteration (Fig. 4a) in CI, CM, and CR chondrites[10,16–18] is fully consistent with the astrophysical CPL scenario. So far identified mechanisms which could be potentially involved in the synthesis of isovaline inside meteorite parent bodies have been considered incapable of producing the L-enrichment without the presence of an initial chiral bias[5]. Therefore, a small but systematic L-excess of isovaline such as the one generated by asymmetric photochemistry, would have been crucial for directing amplification processes during aqueous alteration exclusively toward the L-enantiomer.

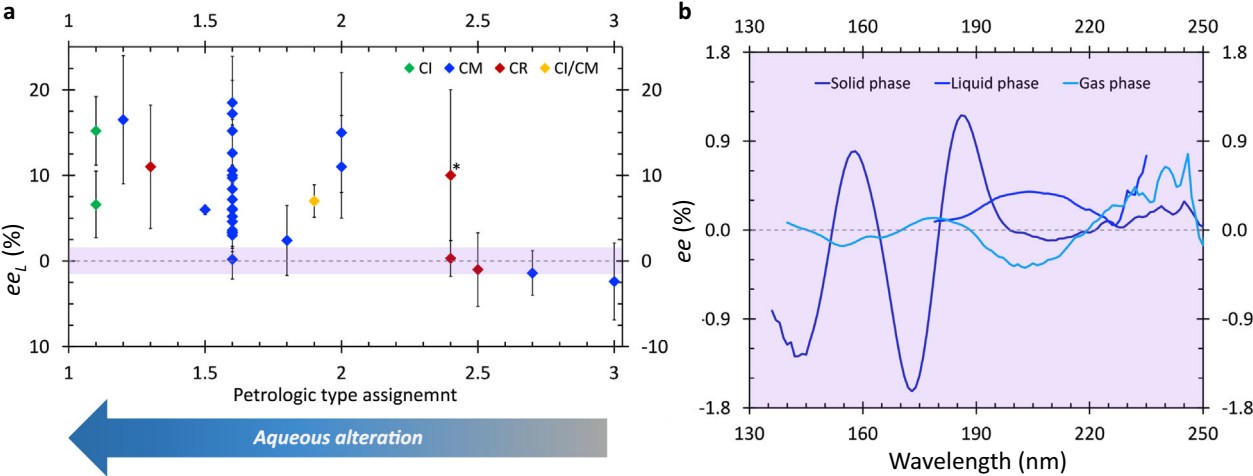

**Fig. 4 | Comparative analysis of isovaline's L-enantiomeric excess in carbonaceous chondrites and inducible by circularly polarized starlight. a** L-enantiomeric excess $ee_L$ of isovaline previously detected in Ivuna-type (CI)[9,56], Mighei-type (CM)[9,10,15,16,44,57–63], CI/CM[18], and Renazzo-type (CR)[9,16,64] carbonaceous chondrites as a function of aqueous alteration (H in OH/$H_2O$ metrics)[63]. Symbols represent the mean and error bars indicate ± standard deviations of $ee_L$ in enantioselective analyses. Only the analyses where the standard deviation of $\%ee_L$ is reported and is equal to or less than 10% are considered (a complete list along with

the number of injections is given in Supplementary Note 5, Supplementary Table 3). The asterisk indicates an anomalous thermally altered CR2.4 chondrite[64,65]. **b** Comparison of the $\%ee$ inducible by ultraviolet circularly polarized light (UV CPL) in solid-, liquid-[33], and gas-phase[36] isovaline calculated from the anisotropy spectra of L-isovaline at the extent of reaction $\xi = 0.9999$ based on the equation reported by Kagan et al.[31]. The upper estimate of the $ee$ inducible solely by interaction with UV CPL is indicated by the purple bar in (**a**), corresponding to the purple area in (**b**). Source data are provided as a Source Data file.

Solid-phase isovaline appears much more efficient in the asymmetry transfer from chiral photons to racemic starting materials as opposed to the liquid-[33] and gas-phase[36] (Fig. 4b). Nevertheless, even in the scenario where stellar CPL interaction would occur predominantly with isovaline adsorbed on interstellar dust grains, the emerging $ee$ would fall below the detection limits of the enantioselective analyses of the most pristine carbonaceous chondrites (purple in Fig. 4). The maximum $|\%ee|$ predicted based on the anisotropy spectrum of L-isovaline in Fig. 2 is ~1.6% at $g_{173} = -0.0035$ and $\xi = 0.9999$. However, given the alternating signs of the anisotropy bands of rather similar intensity, the $ee$ induced by realistic broadband stellar CPL would be considerably lower. Moreover, the relatively high abundance of water-dominated interstellar ices[35] and the presence of gas-dust cycling processes[50] implies that the interactions with CPL would very likely proceed also via much less enantioenriching encounters with a suite of isovaline conformers more closely approximated by the ones found in aqueous solution[33] and gas-phase[36]. Hence, without any amplification purely chiral photon induced $ee$-s potentially present in the most pristine extra-terrestrial samples cannot be observed with the current analytical powers. Further support for the CPL model is the finding that the magnitude of the $ee_L$ of isovaline appears to be inversely related to its total abundance in carbonaceous chondrites[5], since throughout an asymmetric photolysis the optical purity increases at the expense of substrate concentration. Additional studies are needed to elucidate the underlying parent body amplification processes given that the mostly discussed one, based on the crystallization behavior of isovaline during phase transitions, relies on racemization of this α-methyl-amino acid[5,51,52].

In addition to the amplification on Solar System debris, further enantio-enrichment would have been necessary to drive the extra-terrestrial L-excess in amino acids to homochirality on the early Earth. The only well-described example of asymmetric autocatalytic reaction so far is the so-called Soai reaction[53]. Despite lacking prebiotic relevance, the reaction allows to extrapolate for the limits of its yet undiscovered more prebiotically plausible analog. Hawbaker and Blackmond[54] calculated the threshold $ee$ required to direct the Soai reaction with fidelity to be in the range $3.5 \times 10^{-8}$–$3.5 \times 10^{-7}\%$. If we assume the upper limit and the $ee_L$ of isovaline in an aqueously altered

carbonaceous chondrite of ~20%, then asymmetric autocatalysis with Soai-like reaction kinetics would proceed toward the L-enantiomer even if the mass of racemic isovaline at the meteoritic fall site is up to ~$6 \times 10^7$ times greater. This appears generous enough for such amplification to take place, and hence allow L-enriched isovaline to shape the evolution of homochirality of life on Earth.

Finally, besides providing sound evidence for explaining the non-detection of chiral biases of isovaline in the most pristine CI, CM, and CR carbonaceous chondrites as opposed to the aqueously altered ones, the present findings should serve as a guide for the forthcoming search for chiral biosignatures in space. Moreover, they stress the need for the enhancement of detection limits in enantioselective analyses of the most pristine carbonaceous chondrites. The results from the $ee$ analyses of aqueously altered Ryugu and Bennu asteroid return samples, the most intact pieces of the Solar System ever to reach Earth, will provide novel insights into the question of the extra-terrestrial origins of homochirality.

## Methods

### Solid-phase circular dichroism and anisotropy spectroscopy

**Sample preparation.** Enantipure standards of L- and D-isovaline were purchased from Acros Organics and Fisher Scientific, with stated optical purities of 99% and 97%, respectively, and were used without any further purification. Standard solutions of both enantiomers with a concentration of 1 g $L^{-1}$ were prepared by dissolving the samples in methanol (Methanol ≥99.8%, AnalaR NORMAPUR® ACS). Thin films of the order of several hundreds of nm of L- and D-isovaline were produced by drop-casting. 20 μL of a standard solution was dropped on a VUV grade $CaF_2$ window (Crystran) which was preheated to 40–60 °C. Immediately afterwards, up to 60 μL of methanol was added to help distribute isovaline across the window surface by precession motion of the heating system. The heating was applied to aid faster evaporation of methanol and hence achieve homogeneous distribution of isovaline in the central ~5 mm diameter ring on the window. A homemade heating system was built for this experiment consisting of an aluminum block placed in between two Peltier elements connected to a controller. The temperature was monitored by a K-type thermocouple inserted into a tight fit hole in the aluminum block.

**Spectroscopy.** VUV/UV anisotropy spectra of L- and D-isovaline thin films were recorded using the AU-CD beam line of the ASTRID2 synchrotron storage ring facility, Aarhus University, Denmark (Fig. 1b)[21,38]. Briefly, linearly polarized monochromatic VUV/UV (130–330 nm) synchrotron radiation was converted into 50 kHz alternating right- and left-circularly polarized light by a $CaF_2$ photoelastic modulator (PEM). After passing through the thin film deposited on a $CaF_2$ window the transmitted radiation was detected using a VUV enhanced photomultiplier tube. Differential absorption and absorption spectra were measured simultaneously through the conversion of the high voltage applied to the photomultiplier tube as described in ref. 45. To assess potential contributions of linear dichroism and linear birefringence to the CD spectra, we checked for the invariance of the CD signal upon rotation (0°, 90°, 180°, and 270°) of the samples around the axis of the incident synchrotron radiation. Analogous measurements were taken for clean $CaF_2$ windows to separate out any possible contributions from the window and other components of the spectroscopic set-up[42,55]. To confirm the positions of zero-crossings, extrema, and $g$ values, 15 separate thin films were prepared and used for recording the anisotropy spectra for the L-enantiomer and nine for the D-enantiomer. The final data set presented in Fig. 2 was obtained by averaging three anisotropy spectra for the D- and three anisotropy spectra for the L-enantiomer. The main selection criterion for the final data set was the level of scattering. More information on the data treatment procedure as well as raw CD and absorption spectra can be found in the Supplementary Information.

## Asymmetric photolysis experiment

**Sample preparation.** A racemic standard of DL-isovaline (AChemBlock, purity 98%) was purchased from Sigma-Aldrich. Homogeneous racemic thin films with an exact diameter of 3 mm using specially manufactured Teflon masks and a controlled thickness of ~500 nm were produced by deposition of sublimated racemic DL-isovaline powder on $CaF_2$ (Crystran) windows in a homebuilt high vacuum (<$10^{-5}$ mbar) sublimation-deposition chamber (Fig. 1c). Sublimation of DL-isovaline was achieved by resistively heating an aluminum cylinder surrounding the glass sample reservoir to a temperature of 120–129 °C. The rate of deposition on $CaF_2$ windows was checked every 10 min by a 6 MHz quartz crystal microbalance (QCM, Inficon) and controlled by adjusting the temperature of the heating element. For each CPL irradiation experiment, a set of two racemic isovaline films was prepared, i.e. one for irradiation and the other one was left as a non-irradiated reference.

**CPL irradiation at 192 nm.** For asymmetric photolysis experiments on racemic thin films of isovaline a new tunable laser set-up (Fig. 1d) was built at the Institut de Chimie de Nice, Université Côte d'Azur, France. The 192 nm vertically polarized laser radiation was produced using a diode pumped Ekspla NT230 Series tunable laser. The DUV laser pulses (192–210 nm) are generated as sum frequency signal from a 1064 nm pump beam (Nd:YAG, ~3 ns pulse width, 50 Hz repetition rate) and the second harmonic (210–299.5 nm) of the optical parametric oscillator signal (OPO, 405–2600 nm). Linearly polarized laser radiation was reflected by a UV enhanced mirror and was subsequently circularly polarized by a Soleil-Babinet compensator (B. Halle Nachfl. GMBH). The 192 nm circularly polarized laser radiation (~$9 \times 10^4$ W cm$^{-2}$) then photolyzed a ~500 nm thick racemic isovaline film deposited on a $CaF_2$ window and the irradiation progress was monitored by a power meter (Thorlabs, S425C). The $CaF_2$ window continuously moved and was periodically turned by 60° in the xy plane, i.e. perpendicular to the laser beam, to compensate for laser beam imperfections thus allowing for homogeneous sample irradiation. The Soleil-Babinet compensator was calibrated using a Rochon polarizer, and the degree of polarization at 192 nm was ≥97%.

**Analytical procedure.** All glassware used during the analytical procedure was flushed 12 times with Milli-Q water (Milli-Q Direct 8 apparatus,

18.2 MΩ cm at 25 °C, <2ppb total organic carbon), 12 times with ethanol (TechniSolv, purity 96%), 12 times with Milli-Q water, and subsequently heated at 500 °C for 5 h. Polytetrafluoroethylene-lined lids (Merck) were washed in an analogous way. Eppendorf pipette tips, GC vials, inserts and caps (Agilent Technologies) were used without further cleaning. Irradiated DL-isovaline films were extracted from their $CaF_2$ windows with $8 \times 20$ μL of Milli-Q water and transferred into conical reaction vials (1 mL, V-Vial® Wheaton). The solutions were dried under a gentle stream of dry nitrogen and isovaline enantiomers were converted to their N-trifluoroacetyl-O-methyl ester derivatives for subsequent multidimensional gas-chromatography coupled to reflectron time-of-flight mass spectrometry (GC×GC–TOF-MS) analysis. First, 200 μL of methanol (Sigma Aldrich, purity 99.8%)/acetyl chloride (Sigma Aldrich, purity ≥99.0%) solution (4:1, $v/v$) was added to the reaction vials, the reaction mixture was vigorously stirred using a vortex (Stuart™ Scientific SA8) for about 10 s and heated at 110 °C for 1 h. The mixture was then allowed to cool to room temperature for 10 min and dried under a stream of dry nitrogen for 20 min. The solution was not fully dried to avoid potential analyte loss. Subsequently, 200 μL of dichloromethane (Sigma Aldrich, purity ≥99.9%)/trifluoroacetic acid anhydride (Sigma Aldrich, purity ≥99.0%) solution (4:1, $v/v$) was added, the reaction mixture was stirred for about 10 s using the vortex and heated at 100 °C for 20 min. The solution was then fully dried under a gentle stream of dry nitrogen. Finally, the residue was dissolved in 30 μL of $10^{-6}$ M methyl myristate (internal standard, Sigma Aldrich, purity ≥99.0%) in chloroform (Sigma Aldrich, purity ≥99.9%) and transferred to 1 mL GC vials equipped with 100 μL inserts. Aliquots of 1 μL were injected in splitless mode for GC×GC–TOF-MS analysis. To match the signal intensities of $m/z$ 168 fragment ions of the irradiated samples, the non-irradiated counterparts were diluted using MilliQ water and derivatized in an analogous way. The only difference was adding of 50 μL of $10^{-6}$ M methyl myristate in chloroform instead of 30 μL to potentially allow for more replicate injections. This was compromised for the irradiated samples to achieve higher concentrations with limited sample amounts, and consequently higher signal-to-noise ratios ($S/N$)[48]. To exclude any potential instrumental artefacts over the course of the measurements, the irradiated and non-irradiated samples were injected each nine times in a zig-zag mode, i.e. three injections of the irradiated sample followed by a chloroform blank to avoid any potential cross-contamination, then three injections of the diluted non-irradiated counterpart, etc. Where available, additional injections of the non-irradiated sample were done (Table 1).

The enantioselective GC×GC–TOF-MS analyses were carried out using a Pegasus BT 4D instrument coupled to a reflectron TOF-MS (Leco corp.) operating at 150 Hz storage rate, with a 50–500 amu mass range, a microchannel plate detector voltage of ~2–2.1 kV, and a solvent delay of 8 min. The injector and ion source temperatures were kept at 230 °C and the transfer line at 240 °C. The column set consisted of a Lipodex E [octakis(3-O-butanoyl-2,6-di-O-n-pentyl)-γ-cyclodextrin] primary column (24.86 m length × 0.25 mm inner diameter, Macherey-Nagel) coupled to a DB Wax (polyethylene glycol) secondary column (1.4 m length × 0.25 mm inner diameter, 0.1 μm film thickness, Agilent) via a liquid nitrogen jet-based thermal modulator. Helium was used as a carrier gas at a constant flow of 1 mL min$^{-1}$. The primary oven was operated as follows: 30 °C for 1 min, 10 °C min$^{-1}$ to 90 °C, 90 °C for 14 min, 15 °C min$^{-1}$ to 190 °C, 190 °C for 5 min. The secondary oven operated with a constant positive temperature offset of 10 °C, the thermal modulator hot jets temperature offset was set at 15 °C and the modulation period was 4 s. The data were processed using Leco corp. ChromaTOF® software. Representative GC×GC–TOF-MS $m/z$ 168 ion chromatograms are shown in Fig. 3.

## Enantiomeric excess quantification and statistical analysis

L-enantiomeric excess $\%ee_{L\_x}$ of irradiated ($x$ = irr) and corresponding non-irradiated ($x$ = non-irr) samples was determined as the average of

$n_x$%$ee_L$ values (Supplementary Table 1), which were calculated for each individual injection as follows

$$\%ee_L = \frac{A_L - A_D}{A_L + A_D} \times 100\% \qquad (1)$$

where $A_L$ and $A_D$ are $m/z$ 168[48] ion peak counts corresponding to the L- and D-enantiomers of isovaline in single ion chromatogram. It should be noted that for set *III*, the less intense $m/z$ 114[48] fragment ion was used instead for both irradiated and non-irradiated samples in order to avoid any potential artefacts in $ee$ determination due to a peak eluting close to the D-enantiomer of isovaline in the $m/z$ 168 ion chromatogram of the non-irradiated sample.

The CPL induced %$ee_L$ values reported in Table 1 were calculated as follows:

$$\%ee_L = \%ee_{L\_irr} - \%ee_{L\_non\text{-}irr} \qquad (2)$$

and the standard deviation SD was calculated based on the error propagation formula

$$SD = \sqrt{SD_{irr}{}^2 + SD_{non\text{-}irr}{}^2} \qquad (3)$$

where $SD_{irr}$ and $SD_{non\text{-}irr}$ (Supplementary Table 1) are the standard deviations of %$ee_{L\_irr}$ and %$ee_{L\_non\text{-}irr}$, respectively.

A two-sample Student's $t$ test (homoscedastic) was performed in MS Excel to confirm that the differences in %$ee_{L\_irr}$ and %$ee_{L\_non\text{-}irr}$ for individual sample sets are statistically significant. The corresponding $p$ values (two-tailed) are stated in Table 1, and $t$ values and degrees of freedom are in Supplementary Table 1. The absolute %$ee_{L\_non\text{-}irr}$ values measured by GC×GC–TOF-MS (Supplementary Table 1) vary between different sample sets due to (i) the differences in the absolute DL-isovaline concentrations analyzed[48] as well as (ii) changes in peak shapes over the long experimental timescales of 6.5 months and the resulting negative impact of the MeOH/TFAA derivatization byproduct trifluoroacetic acid on the Lipodex E stationary phase. The $ee$ measured by GC×GC–TOF-MS of a set of two diluted non-irradiated films was checked to be the same within statistical uncertainties (Supplementary Table 2).

## Data availability

The authors declare that all data supporting this study are available within the main text and Supplementary Information file. A detailed circular dichroism/anisotropy spectroscopy data treatment procedure, comparison of the anisotropy spectra with our previous study[38], analysis of scanning electron microscopy images, and complementary information on multidimensional gas-chromatography coupled to reflectron time-of-flight mass-spectrometry analyses as well as Supplementary Figs. 1–5 and Supplementary Tables 1–3 can be found in the Supplementary Information file. Raw data of the individual circular dichroism and anisotropy data are provided in the Supplementary Information file. Source data are provided with this paper.

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

## Acknowledgements

This work was funded by the European Research Council under the European Union's Horizon 2020 research and innovation program under grant agreement 804144 (C.M.). Further funding was provided by the project CALIPSOplus, under Grant Agreement 730872 from the EU Framework Program for Research and Innovation HORIZON 2020 (S.V.H., N.C.J., C.M.). J.B. is supported by a postdoctoral fellowship from

the National Centre for Space Studies, CNES (J.B.). We thank F. Orange from the University Côte d'Azur's "Microscopie Imagerie Côte d'Azur" for imaging support and N. Bellouguet for supporting the preparation of isovaline films for SEM imaging.

## Author contributions

C.M. conceived and designed the experiments. J.B., N.C.J., S.V.H., and C.M. performed the spectroscopic investigations. N.C.J., S.V.H., and C.M. conceived and built the tunable laser set-up. J.B. performed the asymmetric photolysis experiments and GC×GC–TOF-MS analyses. J.B. and C.M. recorded SEM images and J.B. analyzed the SEM images. J.T. performed the DFT calculations. J.B. and C.M. wrote the manuscript and Supplementary Information. All authors reviewed and edited the manuscript and Supplementary Information.

## Competing interests

The authors declare no competing interests.
