## [Peer Review File · Nature Communications]

Uncovering the chiral bias of meteoritic isovaline through asymmetric photochemistryREVIEWER COMMENTS

Reviewer #1 (Remarks to the Author):

The present manuscript by Bockova et al describes experiments and calculations to determine the extent to which UV-CPL could drive enantiomeric excesses observed in carbonaceous chondrites, which could have contributed to the origins of chirality on Earth. The work presented is important, and the conclusions are generally well-supported, though in some cases may be a little bit overstated. I am happy to recommend the manuscript for publication but have some suggestions that I think would improve the manuscript.

Line 37 – isovaline-bearing peptides are not “rare”, though they generally contain only D-isovaline or racemic isovaline. See Bruckner et al 2009 *Chemistry and Biodiversity*, “Aib and Iva in the biosphere: neither rare nor necessarily extraterrestrial.” The observation that meteoritic L-excesses of isovaline are actually opposite of the predominant chirality of this amino acid when it is used in biology, appears at least ironic, if not contradictory to the argument that the L-excesses observed for isovaline are related to the origins of L-homochirality in translated proteins, let alone D-sugars in nucleic acids and carbohydrates.

Lines 202-205: While I understand the rationale for choosing to irradiate at 192 nm, how would the results be changed by full spectrum UV-CPL from a reasonable star source, including the relative flux of photons over the range of UV light that would have an asymmetric effect? Can the authors address whether the expected enantiomeric excesses would be greater or lesser than the value? I see there's some text on this in the discussion, so I think just directing the reader that this will be addressed later would be sufficient

Lines 229-230: CH and CB chondrites also show L-isovaline excesses of 10-20% (e.g. Burton et al. *Meteoritics & Planetary Science* 2013, “Extraterrestrial amino acids identified in metal-rich CH and CB carbonaceous chondrites from Antarctica).

Line 254 - The crystallization behavior of isovaline (preferential formation of enantiopure crystals from a racemic solution) was characterized in Butcher et al. 2013 *Acta Crystallographica Section E*, “Isovaline monohydrate”. This may be useful to cite, in addition to references 5 and 51, neither of which contains the actual crystallographic data.

Figure 4: Can you at least put in the supplementary information which meteorites are actually included in the figure (i.e., which have standard deviations less than 1.5 times the average ee)? MIL 090001 may meet this criteria as well, as an additional CR chondrite (Aponte et al. 2020 *Meteoritics & Planetary Science* “Analysis of amino acids, hydroxy acids, and amines in CR chondrites).

Reviewer #2 (Remarks to the Author):

In their study "Uncovering the chiral bias of meteoritic isovaline through asymmetric photochemistry", Meinert and co-workers present the solid-state circular dichroism of isovaline (recorded at the ASTRID2 synchrotron storage ring facility in Aarhus) and the results from irradiation experiments with circular polarized light. The solid-state CD spectra and the therefrom derived anisotropy spectra (g-factor spectra) were found to be similar to those reported for other amino acids by the same group. The photolysis experiments (10 hours irradiation with analysis/quantification based on GCxGC-TOF MS) show that some enantiomeric excess can indeed be obtained (about 1% as predicted by computations). While the experiments are very well designed and carried out and the results are certainly interesting and sound, the manuscript reads a bit far away from the actually investigated molecule. In fact, many paragraphs in the discussion section are really rather general and it appears as if isovaline could easily be replaced by any other of the previously investigated amino acids. The writing style is occasionally also less "broader audience"-friendly; for instance, "CM, CI, and CR chondrites" is not further explained (until the end), what is meant with "aqueous alteration", what is "CPL-helicity dependent enantiomeric excesses", what is the Kagan equation (it is often referenced, never shown), ... I fear, despite having cool experiments, the overall manuscript and the data described in it, is better suited in a more specialized journal.

Reviewer's #1 report

The present manuscript by Bockova et al describes experiments and calculations to determine the extent to which UV-CPL could drive enantiomeric excesses observed in carbonaceous chondrites, which could have contributed to the origins of chirality on Earth. The work presented is important, and the conclusions are generally well-supported, though in some cases may be a little bit overstated. I am happy to recommend the manuscript for publication but have some suggestions that I think would improve the manuscript.

Response: Thank you for your overall positive evaluation of the manuscript and your insightful remarks that helped refine the manuscript.

Line 37 – isovaline-bearing peptides are not “rare”, though they generally contain only D-isovaline or racemic isovaline. See Bruckner et al 2009 Chemistry and Biodiversity, “Aib and Iva in the biosphere: neither rare nor necessarily extraterrestrial.” The observation that meteoritic L-excesses of isovaline are actually opposite of the predominant chirality of this amino acid when it is used in biology, appears at least ironic, if not contradictory to the argument that the L-excesses observed for isovaline are related to the origins of L-homochirality in translated proteins, let alone D-sugars in nucleic acids and carbohydrates.

Response: Thank you for pointing this out. We have removed the word “rare” from the following sentence in line 37: “Given that isovaline is mostly absent in the Earth’s biosphere except for a few ~~rare~~-fungal peptides containing predominantly the D-form¹¹, the molecule represents a robust test case in the quest for chiral excess in extra-terrestrial samples.”

The present manuscript primarily focuses on investigating and explaining the ee of isovaline that can be triggered by stellar CPL in the context of its distinct chiral excesses detected in CI, CM, and CR carbonaceous chondrites. Given that isovaline is found in fungal peptides predominantly in the D-form or in racemic ratios, potential contaminations of meteoritic samples cannot explain the large L-excess of isovaline detected in several carbonaceous chondrites. Therefore, isovaline’s large L-ee in meteorites remain a robust indication of extra-terrestrial origins of symmetry breaking, with the CPL scenario being a strong candidate for its origin.

Even though isovaline appears in fungi, it is not common to all life forms on Earth, but rather a unique trait of this specific group of organisms. Moreover, the fact that isovaline can appear either racemic or as the D-enantiomer in peptaibiotics indicates that the chirality of this species is not crucial. In general, this is not the case for α -amino acids in proteins or sugars in nucleic acids, where the opposite chirality would be detrimental. This suggests that isovaline likely got incorporated during the evolution rather than having its origins in the symmetry breaking event, which would have systematically directed the omnipresent homochirality of amino acids and sugars. The presence of isovaline in peptaibiotics and its chiral preference in contrast to the L-excess found in carbonaceous chondrites is certainly an interesting scientific question to investigate, but obviously beyond the scope of the present manuscript.

Lines 202-205: While I understand the rationale for choosing to irradiate at 192 nm, how would the results be changed by full spectrum UV-CPL from a reasonable star source, including the relative flux of photons over the range of UV light that would have an asymmetric effect? Can the authors address whether the expected enantiomeric excesses would be greater or lesser than the value? I see there’s some text on this in the discussion, so I think just directing the reader that this will be addressed later would be sufficient

Response: Thank you for this remark. We have added the following sentence to the line 205: “While this result further supports the accuracy of the anisotropy spectra in **Fig. 2**, the effect of asymmetric

photolysis by more astrophysically relevant broadband stellar CPL and the implications for the CPL scenario are elaborated in the Discussion section.”

Lines 229-230: CH and CB chondrites also show L-isovaline excesses of 10-20% (e.g. Burton et al. Meteoritics & Planetary Science 2013, “Extraterrestrial amino acids identified in metal-rich CH and CB carbonaceous chondrites from Antarctica).

Response: We have now added the corresponding reference to **Supplementary Table 3** in the Supplementary Information file. Lines 229–230 discuss the change in the *ee* of isovaline with the extent of aqueous alteration in CI, CM, and CR chondrites. Another process may be responsible for enhanced *ee*-s of isovaline in CH and CB chondrites due to their distinct parent body histories, the investigation of which is beyond the scope of this manuscript.

Line 254 - The crystallization behavior of isovaline (preferential formation of enantiopure crystals from a racemic solution) was characterized in Butcher et al. 2013 Acta Crystallographica Section E, “Isovaline monohydrate”. This may be useful to cite, in addition to references 5 and 51, neither of which contains the actual crystallographic data.

Response: We have included this reference in the corresponding sentence in the manuscript (line 254).

Figure 4: Can you at least put in the supplementary information which meteorites are actually included in the figure (i.e., which have standard deviations less than 1.5 times the average ee)? MIL 090001 may meet this criterion as well, as an additional CR chondrite (Aponte et al. 2020 Meteoritics & Planetary Science “Analysis of amino acids, hydroxy acids, and amines in CR chondrites).

Response: Thank you for highlighting this reference, we have added it to **Fig. 4** of the main manuscript. We have also added to the **Supplementary table 3**, to the best of our knowledge, all detections of *ee* of isovaline in carbonaceous chondrites.

We have also corrected the criterion for the *ee*-s which are plotted in **Fig. 4** in the main manuscript to “standard deviation being less than 10%” instead of “less than 1.5-times the average *ee*” which would be incorrectly restrictive for %*ee*-s close to 0%. In addition to Aponte et al., 2020, only a single reference for a CI chondrite “Burton, A. S., Grunsfeld, S., Elsila, J. E., Glavin, D. P. & Dworkin, J. P. The effects of parent-body hydrothermal heating on amino acid abundances in CI-like chondrites. *Polar Sci.* **8**, 255–263 (2014)” was added to **Fig. 4**. The caption was modified accordingly: “**Fig. 4 Enantiomeric excess (*ee*) of isovaline inducible by UV CPL within detection uncertainties of the enantioselective analyses of the most pristine CI, CM, and CR carbonaceous chondrites. a** L-enantiomeric excess %*ee*_L of isovaline previously detected in CI^{9,53}, CM^{9,10,15,16,44,54–60}, CI/CM¹⁸, and CR^{9,16,61} carbonaceous chondrites as a function of aqueous alteration (H in OH/H₂O metrics⁵⁵). Only the analyses where the standard deviation of %*ee*_L is reported and is less than 10% are plotted (a complete list is in Supplementary Note 5, **Supplementary Table 3**). *Described as an anomalous thermally altered CR2.4 chondrite^{61,62}. **b** Comparison of the %*ee* inducible by CPL in solid-, liquid-³³, and gas-phase³⁶ isovaline calculated from the anisotropy spectra of L-isovaline at the extent of reaction $\xi = 0.9999$ based on the equation reported by Kagan et al.³¹ The upper estimate of the *ee* inducible solely by interaction with CPL is indicated by the purple bar in **a**.”

Reviewer's #2 reports

In their study "Uncovering the chiral bias of meteoritic isovaline through asymmetric photochemistry", Meinert and co-workers present the solid-state circular dichroism of isovaline (recorded at the ASTRID2 synchrotron storage ring facility in Aarhus) and the results from irradiation experiments with circular polarized light. The solid-state CD spectra and the therefrom derived anisotropy spectra (g-factor spectra) were found to be similar to those reported for other amino acids by the same group. The photolysis experiments (10 hours irradiation with analysis/quantification based on GCxGC-TOF MS) show that some enantiomeric excess can indeed be obtained (about 1% as predicted by computations). While the experiments are very well designed and carried out and the results are certainly interesting and sound, the manuscripts reads a bit far away from the actually investigated molecule. In fact, many paragraphs in the discussion section are really rather general and it appears as if isovaline could easily be replaced by any other of the previously investigated amino acids.

Response: Thank you for acknowledging the experimentation and quality of the results presented in the manuscript. We would like to highlight the uniqueness of isovaline for the present study and in general, in the origin-of-life research. As described in the manuscript (lines 34–44), isovaline is a non-proteinogenic amino acid, found only in specific life forms on Earth, and if present in the biosphere, it is either racemic or enriched in the D-enantiomer. Due to isovaline's α -methyl group, it is much more resistant to racemization over geological timescales than α -hydrogen amino acids, and thus has greater potential to preserve its chiral bias. Moreover, the chromatographic analyses of extraterrestrial samples are challenging due to small sample quantities, distorting matrix effects, large diversity of organics present in meteorites potentially leading to co-elutions, and potential terrestrial contaminations. Therefore, detection of subtle L-ee-s of proteinogenic amino acids in meteorites may be ambiguous. However, this is clearly not the case for isovaline, which stands out amongst other amino acids detected in the carbonaceous chondrites. Its significant L-excess of up to about 20% in several chondrites can be with the current knowledge unambiguously marked as indigenous to meteorites. Finally, the enantiomeric excess of isovaline changing with the extent of aqueous alteration in CI, CM, and CR chondrites has been a subject of scientific discussions over the last two decades. Therefore, we disagree with the reviewer that isovaline could be easily replaced by any other previously investigated amino acid.

In the Discussion section, drawing on the experimental results while considering more realistic astrophysical conditions, we discuss how the present results could explain the results of enantioselective analyses of isovaline in carbonaceous chondrites. Most of the discussion section is, therefore, strongly focused on isovaline:

- The first paragraph highlights the uniqueness of isovaline and its potential importance for shaping the evolution of homochirality of other amino acids and sugars. Thence, the paragraph is not general, but rather strongly related to the specificity of isovaline.
- The second paragraph is focused on explaining what would be the net effect of stellar broadband CPL acting on isovaline and how this could explain its enantiomeric excess which varies with the extent of aqueous alteration in carbonaceous chondrites, i.e. again the paragraph is strongly related to the specificity of isovaline.
- The third paragraph is focused on assessing whether isovaline's ee from meteorites could have been significant enough for affecting the evolution of homochirality of other species. The paragraph is, therefore, again based on the high ee of isovaline detected in several carbonaceous chondrites and the fact that isovaline was shown to be able to affect the chirality of other species.
- The fourth paragraph is the only one where we draw more general conclusions and implications. Still, isovaline is one of the major targets in enantioselective analyses of meteorites and return samples. Therefore, the outcomes on isovaline presented in the manuscript will be very important and useful for directing the future analytical procedures applied to these precious samples.

The writing style is occasionally also less "broader audience"-friendly; for instance, "CM, CI, and CR chondrites" is not further explained (until the end), what is meant with "aqueous alteration", what is "CPL-helicity dependent enantiomeric excesses", what is the Kagan equation (it is often referenced, never shown), ... I fear, despite having cool experiments, the overall manuscript and the data described in it, is better suited in a more specialized journal.

Response: Thank you for the suggestion to clarify some terms so that the manuscript is more easily accessible to even broader readership. In this context we have added the following explanations (underlined) in the text:

Line 34–36: "The non-proteinogenic amino acid isovaline (**Fig. 1a**) stands out amongst most extra-terrestrial chiral organics due to its presence in large L-enantiomeric excess (ee_L) of up to about 20% in a number of carbon-rich meteorites, so called carbonaceous chondrites^{8–10}."

Line 42–45: "Interestingly, the magnitude of ee_L of isovaline shows a positive correlation with the extent of aqueous alteration in CI, CM, and CR chondrites (classes petrologically and compositionally similar to Mighei, Ivuna, and Renazzo meteorites, respectively)^{10,16–18}, while it appears inversely related to its overall abundance^{5,16,19}, and demonstrates yet another unique property of meteorites."

Line 45–49: "Even though the impact of aqueous alteration caused by melted water ice on the inorganic mineral content of parent bodies is rather well understood, the underlying mechanisms that explain how these geological processes would have affected the ee are not yet fully elucidated, despite their significant relevance for chiral amplification."

The meaning of "CPL-helicity dependent enantiomeric excesses" is explained in detail in the Introduction section, lines 70–75, and we prefer to use this succinct form in the abstract to maintain the brevity and flow of the text. Finally, the term says what it reads, i.e. enantiomeric excesses which are dependent on the helicity of circularly polarized light.

We have added the Kagan's equation to the Supplementary Information file:

"Supplementary Note 1: Kagan's equation¹

Kagan's equation¹ describes how the enantiomeric excess ee_L of a racemic mixture in an asymmetric photolysis evolves with the extent of reaction ξ :

$$\xi = 1 - \frac{1}{2} \left[\left(\frac{1 + ee_L}{1 - ee_L} \right)^{\frac{1}{2} - \frac{1}{g_L}} + \left(\frac{1 + ee_L}{1 - ee_L} \right)^{-\frac{1}{2} - \frac{1}{g_L}} \right] \quad (S1)$$

where g_L is the anisotropy factor corresponding to the L-enantiomer and depends on the wavelength of the circularly polarized light."

A reference to Supplementary Note 1 was added to the main manuscript (line 79).

In the following, we underline the motivation and impact of the present study to emphasise the broad interest of the manuscript for the readership of *Nature Communications*. Chirality is omnipresent in nature and homochirality of amino acids in proteins and sugars in nucleic acids is a key biosignature common to all life forms on Earth. Despite its vital importance, the origins of biological homochirality remain unknown. Answering this question is a key step towards elucidating how life began, which is one of the greatest questions of humanity, as well as finding potential extant/extinct life in space. Given the L-enrichments of amino acids in meteorites, the astrophysical circularly polarised light scenario appears very promising for explaining the origin of symmetry breaking in space. To validate the scenario, comparisons of the net effect of broadband CPL with the results of enantioselective analyses of extra-terrestrial samples are necessary. For reasons already mentioned earlier (see above and the Introduction section of the main manuscript, lines 34–44), isovaline stands out amongst other chiral organics detected in meteorites and hence is a strategic molecule for such studies. The present manuscript therefore discusses the results of chiroptical spectroscopy and asymmetric

photochemistry of isovaline in light of the astrophysical CPL scenario and origins of homochirality. Furthermore, it provides the first experimentally supported explanation for the contrasting results of the enantioselective analyses of isovaline in CI, CM, and CR carbonaceous chondrites, which have been discussed extensively in the origin of life community over the last two decades. Finally, the outcomes can be also considered as an important guide for astrobiologists and scientists analysing meteorites to improve their detection uncertainties as well for preparing the analytical procedures for the sample return missions, notably Hayabusa 2 and OSIRIS-REx. This is crucial for reaching a common goal of elucidating the chiral force responsible for symmetry breaking and consequently the origin of life on Earth. Therefore, in view of this and given the extra clarification of the few specific terms we added to the manuscript and SI, we are confident that the manuscript is of great interest and well accessible for the broad readership of *Nature Communications*.

REVIEWERS' COMMENTS

Reviewer #1 (Remarks to the Author):

The authors have satisfactorily addressed the reviewer comments and I am happy to recommend the manuscript for publication.

Reviewer #2 (Remarks to the Author):

The authors have addressed my concerns and I have no scientific objections against publication.